# Cigarette Smoking Blunts Exercise-Induced Heart Rate Response among Young Adult Male Smokers

**DOI:** 10.3390/ijerph16061032

**Published:** 2019-03-21

**Authors:** Sri Sumartiningsih, Hsin-Fu Lin, Jung-Charng Lin

**Affiliations:** 1Department of Sports Science, Semarang State University (UNNES), Jalan Raya Gunung Pati, Semarang, Central Java 50229, Indonesia; 2Graduate Institute of Sports Coaching Science, Chinese Culture University, No.55 Hua Guang Rd Shilin District, Taipei 11114, Taiwan; 3Department of Athletics, National Taiwan University, No.1, section 4, Roosevelt Rd, Taipei 10617, Taiwan; hsinfu@ntu.edu.tw

**Keywords:** e-cigarette, tobacco cigarette, exercise performance, heart rate response, heart rate variability

## Abstract

This study aimed to examine the exercise-induced heart rate response (HRR) and heart rate variability (HRV) in subjects caused by inhaling smoke from tobacco cigarettes (TC) and aerosolized vapor from electronic nicotine dispensing systems (ENDS) (commonly referred to as e-cigarettes (EC)). A randomized crossover study recruited 24 young adult male smokers with an average age of 23 years and with a smoking habit of at least two years. Heart rate response was recorded after a maximal multistage shuttle 20 m run test (MMST) under three different levels of nicotine: Control 0 mg nicotine of EC (C), 3 mg nicotine of EC (3EC), and 3 mg nicotine of TC (3TC). HRV was evaluated based on the beat-to-beat time interval during the running test. The results showed no statistically significant differences in the run time to exhaustion under the three conditions (C: 398 ± 151 s; 3EC: 399 ± 160 s; 3TC: 381 ± 150 s). Exercise-induced HRR was significantly attenuated under the TC condition (*p* < 0.05). Intriguingly, the HRV standard deviation of normal-to-normal intervals (SDNN) during exercise significantly increased under 3EC and 3TC. The results showed that a significant acute autonomic cardiac modulation during exercise is induced by an acute episode of EC and TC smoking.

## 1. Introduction

Globally, over 1.1 million people smoke tobacco cigarettes (TC) and this number is still increasing [1]. The high rate of health complications related to cigarette smoking is also increasing, with more than one out of ten cases of cardiovascular deaths (54% of all deaths) being caused by smoking cigarettes [2]. Cardiovascular death caused by smoking accounts for 28% and 13% of deaths in males and females between 35–69 years old, respectively [3].

TC smoking has an effect on physiological response and exercise performance. Smokers tend to have a higher resting heart rate (HR) compared to non-smokers, before and after a maximum Bruce treadmill test (298 healthy men and women aged between 20–29 years old). Heart rate during recovery was lower in both genders for smokers compared to non-smokers [4,5]. Acute exposure of waterpipe tobacco smoking caused a decrease in lung function and exercise capacity in healthy young male smokers between 18–26 years old [6].

One of the major risk factors of smoking is related to atherosclerosis. However, its effects on heart rate responses (HRR), particularly the initiation and progression of cardiac systolic and diastolic dysfunction, remain unknown [7]. Based on a test conducted on male mice, chronic smoking (48 min/day, 5 days/week for 16 or 32 weeks) showed a significant decrease in body weight, raised blood pressure (BP), impaired cardiac function, and accompanied by an increase in the number of white blood cells, the rate of nitric oxide (NO) decay in the blood, and endothelial dysfunction [8]. In another study, five minutes of smoking caused a significant effect on the BP, HR, vessel dilation ratio, aortic dispensability, and the aortic stiffness index in a sample of 34 smokers aged 21–35 years old [9].

In addition, smoking also reduced the level of endurance in teenager smokers (18–19 years old). The subjects recorded slower times in 1500 m and 10,000 m runs [10]. Acute effects of smoking two cigarettes also increased HR during rest and exercise [11].

Nowadays, a number of tools for smoking cessation are offered on the market. Electronic nicotine delivery systems (ENDS), commonly referred as e-cigarettes (EC), and electronic non-nicotine delivery systems (ENNDS) have been considered as a safety measure and a tool to help people stop smoking tobacco [12,13]. Manufacturers have claimed that these delivery systems help to reduce the number of tobacco smokers. These delivery systems involve heating up a solution, often referred to as e-liquid or e-juice, to generate an aerosol. The user inhales the aerosol and then exhales it back. Usually, the solutions contain flavorings dissolved into propylene glycol and/or glycerin [14].

Previous studies have found that using EC increases the risk of cardiac arrhythmias and hypertension [15,16]. The acute effect of inhaling EC showed a significant difference between 0 mg (placebo) and 18 mg nicotine (nicotine trial) in terms of the rest, exercise, and peak exercise diastolic blood pressure (DBP) of non-smokers. A resting condition of 40 min of inhalation found that the nicotine trial was higher in DBP but lower in systolic blood pressure (SBP) than the placebo. However, the resting metabolic rate (RMR) and oxygen uptake during peak exercise, commonly known as peak oxygen uptake (VO_2peak_), showed no significant difference between treatments [17].

Due to limited research on the effects of EC, the WHO has encouraged researchers from around the world to evaluate the health effects of ENDS on EC users [18]. Furthermore, the different effects of EC and TC on exercise performance are still unknown. This study aims to investigate the acute effects of different methods of smoking EC and TC, and different dosages of nicotine (0 mg of EC; 3 mg nicotine of EC and TC), on physiological responses and exercise performance. It was hypothesized that the physiological responses and exercise performance of EC and TC users would be the same.

## 2. Materials and Methods

### 2.1. Participants

Initially, 30 young adult male smokers volunteered and were recruited for this study. The participants were on average 23 years old, a height of 1.70 m, weight of 63.25 kg, and body mass index (BMI) of 21.82 kg/m^2^. All were regular smokers, smoking on average 9 TC per day, with a smoking habit of approximately 3.5 years. Participants were required to smoke EC with at least 0 mg of nicotine. Six participants were dropped from the study because they failed to complete all testing sessions.

The inclusion criteria included: (1) a physical examination by a physician, (2) blood pressure in a range below 120/80–130/80 mm Hg, and (3) no other risk factors (e.g., cardiovascular disease and diabetes mellitus). All participants signed a participation consent form before attending the experiment.

### 2.2. Study Design

This study used an experimental design with repeated measures with the same participant and a randomized crossover design (balance order treatment) [17]. Each participant was subjected to three different test sessions held at intervals every three days. For each treatment, the participants were assigned to smoke EC with zero nicotine/ENNDS (C), EC with 3 mg/mL of nicotine/ENDS (3EC), and two tobacco cigarettes (TC) with 1.5 mg nicotine in each (3TC) without knowing the nicotine levels.

The study used a lower concentration of nicotine as a lower level of nicotine helped people quit smoking [19].

### 2.3. Testing Protocol

All participants started their sessions from 9:00 a.m. to 12:00 p.m. Participants were required to refrain from smoking for at least six hours, not to drink coffee and eat at least two hours before arriving at the laboratory. After arrival, they were asked to rest for 5–10 min. Next, HR and BP were measured using a HR monitor (Pulse Oximeter, Elitecth, PT. Sinko Prima Alloy, Surabaya, Indonesia) and automatic sphygmomanometer (Omron Healthcare Co., Ltd., Kyoto, Japan) to obtain the baseline data.

After the medical examination, participants were asked to smoke EC or TC without knowing the dosage of nicotine, HR and BP were measured immediately after smoking. Puffs and smoking duration were calculated during this session. Participants then performed a maximal multistage 20 m of shuttle run test (MMST). Heart rate and BP were measured again immediately after the MMST test. The levels and shuttles of running during the MMST test were recorded to predict the maximal oxygen uptake ($\dot{V}$O_2max_) and to assess the time to exhaustion. 

### 2.4. Heart Rate Variability (HRV)

This study used a Polar RS800X Heart Rate Monitor Polar Electro (Kempele, Finland) to measure the HR and R wave intervals (RR interval) of the participants. The measurement of HRV was conducted while the MMST test was being performed. The generated data from the software (Polar Pro Trainer 5 Software) were transferred and stored as a text file on the computer based compatible software Kubios HRV version 2.2 for analysis. The linear indices obtained in the time domain are as follows: mean RR interval, root mean square of successive differences between adjacent normal RR intervals, and standard deviation of RR intervals. In the frequency domain, the following linear indices were obtained: the VLF, the HF component, the low-frequency (LF) component, and the LF/HF ratio [20].

Heart rate variability was analyzed in the time, frequency, and nonlinear domains. The time domain measured included HR (beats per minute), mean of R-R intervals (mean R-R (ms)), standard deviation of NN interval time series (SDNN (ms)), root mean square of successive differences in the NN intervals (RMSSD (ms)), number of interval differences of successive NN intervals greater than 50 ms (NN50 (number)), and NN50 count divided by the total number of all NN intervals (pNN50 (%)). The frequency domain was characterized by the Fast Fourier Transformation (FFT), in which the low frequency (LF (ms^2^), 0.04–0.15 Hz), high frequency (HF, (ms^2^) 0.15–0.4 Hz), and low frequency-to-high frequency ratio (LF (ms^2^)/HF (ms^2^)) components of the RR interval time series were analyzed. Nonlinear domain measures included standard deviations of the Poincaré plot short-term (SD1 (ms)) and long-term (SD2 (ms)) [21].

### 2.5. A Maximal Multistage 20 m Shuttle Run Test (MMST)

The MMST was designed to evaluate the maximal aerobic power of young to elderly adults. The participants were required to run back and forth on a 20 m course and touch the 20 m line. A sound signal was emitted from a prerecorded tape at the same time. For the MMST, the participants were required to run until they felt exhausted, and the levels and shuttles were then calculated [22].

### 2.6. Time-to-Exhaustion Analyses

Predicted $\dot{V}$O_2max_ was gauged based on the results of the running level and the number of shuttles that could be reached by the participants until they felt exhausted, or when they were late in achieving the next level three times. The running level and the number of shuttles can be used to predict VO_2max_ and time-to-exhaustion [23,24,25]. This study also used the same technique to predict time to exhaustion.

### 2.7. Statistical Analyses

The data were tested for normality. Comparisons were made among the three levels of nicotine and test sessions using a repeated measures ANOVA. Significant changes (*p* < 0.05) were then post hoc analyzed using a paired t-test with Fisher correction for multiple comparisons. Comparisons of the two subgroups for similarities, and during the continuous period, used an unpaired *t*-test. A two-tailed distribution with Fisher correction with probability *p* < 0.05 was considered significant [26]. The results are presented as mean, plus or minus (±) the standard deviation (SD).

### 2.8. Institutional Review

The Institutional Review Board (IRB) of the Faculty of Medicine Diponegoro University, Semarang, Indonesia, approved this study (Reference number: 580/EC/FK-RSDK/IX/2017), as well as the original study protocol. All participants were provided with information on consent for this study.

## 3. Results

Twenty-four young male smokers participated in interviews and anthropometry for this study. The characteristic data of age, height, weight, body mass index (BMI), the history of cigarettes per day, and the duration of smoking are shown in Table 1.

The responses of HR, SBP, and DBP are shown in Table 2. The mean values indicated no significant differences on SBP among the three groups; C, 3EC, and 3TC (F = 0.947, *p* = 0.393). Significant differences were detected between 0EC and 3TC after smoking for HR (*p* = 0.028) and DBP (*p* = 0.017). The changes in HR between groups following exercise were not significantly different (F = 1.864, *p* = 0.163), nor was the HR change following exercise between C and 3TC (*p* = 0.063). Moreover, the change following exercise for SBP (F = 0.453, *p* = 0.638) and DBP (F = 0.947, *p* = 0.393) also showed no significant difference. The treatments of pre-smoking and post-smoking on post-exercise showed a statistically significant difference with time on HR (F = 1183.73, *p* = 0.000) and SBP (F = 212.459, *p* = 0.000). No significant difference was observed for DBP (F = 1.535, *p* = 0.219) (Table 2). The time to exhaustion showed no statistically significant difference in response to the various treatments (C, 3EC and 3TC) (F = 0.110, *p* = 0.890), see Table 3.

The mean variable during exercise, presented in Table 4, found a significant difference of the HRV mean variables during exercise between groups C and 3TC on SDNN (*p* = 0.012), RMSSD (*p* = 0.003), and HR max (*p* = 0.008). However, mean HR (*p* = 0.067), LF (*p* = 0.060), and HF (*p* = 0.060) showed no significant difference, followed by VLF (*p* = 0.837) and power LF/HF (*p* = 0.188). A significant difference between C and 3EC was found on mean SDNN (*p* = 0.011), RMSSD (*p* = 0.048), min HR (*p* = 0.045), and mean HR (*p* = 0.003). The max HR was almost significantly different (*p* = 0.80), followed by VLF (*p* = 0.987), LF (*p* = 0.297), HF (*p* = 0.690), and power LF/HF (*p* = 0.590).

## 4. Discussion

The results revealed that acute smoking of two cigarettes containing 3 mg nicotine (TC) immediately increases HR and reduces DBP compared to EC. Both EC with and without nicotine showed the same effect on HR, SBP and DBP after smoking. The study found no significant differences between smoking without nicotine (C) and with nicotine (3EC and 3TC) on predicted VO_2max_ and time to exhaustion in healthy young adult male smokers. Previous studies conducted with healthy young adult smokers found that smoking 2–3 cigarettes pre-exercise reduced the VO_2max_ by between 4% and 7% [27,28]. This study reported that young male smokers using EC and TC without and with 3 mg nicotine before exercise were able to perform the same in sports.

Acute smoking of 3EC and 3TC before exercise increases autonomic modulation on SDNN and RMSSD in performance. This study confirmed that adaptations to exercise training indicate that HRV analysis is more responsive in the time domain than in the frequency domain [29]. However, this study provided evidence that SDNN, RMSSD, HRmax, and HRmean are more sensitive, with significant differences between control without nicotine and treatment with 3 mg nicotine in EC and TC. The findings of increased SDNN and RMSSD in the 3EC and 3TC groups compared to control during the exercise performance test were interesting as this occurred in healthy young adult smokers. The lowest HRmax and HRmean were observed in the control group due to 3 mg nicotine intake, while smoking increased the HR [4].

This study used acute EC (0 and 3 mg/mL of nicotine), TC (3 mg nicotine/two cigarettes), and utilized the MMST for habitual young adult male smokers. There were no significant differences in physiological responses (SBP and DBP) to the levels of nicotine (C, 3EC, and 3TC) after exercise. HR changes also showed no significant difference between C and 3TC. Time to exhaustion was not significantly different between treatments. 

This study found a significantly different effect on physiological responses at different times (baseline, post-smoking, and post-exercise) on HR and SBP but not on DBP, which was induced by the exercise effect. SBP gradually followed the increase of HR post-exercise. Meanwhile, the DBP in each treatment was not significantly different at pre-smoking, post-smoking, and post-exercise. These findings indicate significant differences in HR and DBP between 0EC and 3TC after smoking. This means that the acute effect of 3 mg nicotine of TC increased HR and DBP immediately after smoking. Furthermore, we also found that during exercise, autonomic cardiac modulation by 3EC and 3TC caused an increase of SDNN and RMSD. Maximum HR during exercise showed that 3TC induced a lower max HR than C.

A previous study conducted on 17 male and female smokers and non-smokers aged 64 years old found no significant difference in the SBP and DBP level [30]. However, both male and female smokers had a slower increase of HR during exercise compared to non-smokers. Slower HR increase during exercise, rest, and maximum performance was frequently associated with younger adult smokers [4]. For both male and female, healthy middle-aged, and elderly smokers (64 years), smoking caused a decrease in the stimulation of the peripheral microvascular response in both the endothelial and smooth muscle cells [30].

An increase in cardiac output, which was correlated with HR and stroke volume, was caused by an increase in metabolism during exercise. During exercise, an increase in HR was influenced by age, resting HR, and exercise load [31]. During exercise with autonomic control, HR increased along with the escalation of sympathetic activity and a reduction of vagal tone. When someone reached max HR (±10 bpm), which was close to their age-predicted HR max, they were considered to be on the right track to enhance his or her HR during exercise [32]. Although the mean HR post exercise in C was higher than in 3EC and 3TC, our findings did not reveal statistically significant differences in regards to post-exercise.

However, in this study, DBP after smoking in 3TC was significantly higher than in 3EC and C. Fogt et al. [17] also discovered that, during resting and exercising, the DBP among 20 young adult non-smokers (avg = 23.1 years) following 20 inhalations of 18 mg/mL nicotine using EC was significantly higher than those inhaling 0 mg/mL nicotine. However, resting SBP was significantly lower for the higher nicotine trial.

Furthermore, we found no significant difference in the mean of the predicted VO_2_max and time to exhaustion among treatments (C, 3EC, and 3TC). The calculation results of time to exhaustion of smokers were similar for EC and TC smoking. Thus, the same effect on physical performance in this study may have been caused by different nicotine intake (0 and 3 mg), different ways of smoking (EC and TC), and different nicotine intake with different dosages. Our results are in concordance with studies by Huie [33] and Druyan et al. [34], where the nicotine intake among the 0EC, 3EC, and 3TC groups showed no significant differences in the predicted $\dot{V}$O_2max_ and time to exhaustion.

Our study reveals that smoking with and without nicotine (ENNDS) before the MMST test showed no difference in the predicted VO_2_max and time to exhaustion for young adult male smokers. In contrast, studies by Klausen et al. [27] on 16 male young smokers reported that smoking during rest and maximum exercise on a Krogh cycle ergometer significantly reduced VO_2max_ by 7% and endurance time by 20%. Nicotine intake during smoking could enhance heat strain and lead to a potential risk of impaired physical performance and injuries [34]. The ingredients in tobacco, such as carbon monoxide (CO), will reduce aerobic performance [27], while tar has some effects on regulating vascular tone by enhancing endothelin-1 and depressing nitric oxide (NO) [35]. 

## 5. Conclusions

This study revealed that the acute effects of different ways of smoking (EC and TC) and nicotine dosage (0 mg, 3 mg) were able to induce significantly different responses in autonomic modulation during exercise.

The strength of this study lies strictly in the selection criteria of the subjects. Our subjects were young healthy adult smokers with smoking status. We also recorded their smoking history (time and frequency). However, a limitation of this study was the absence of a control group consisting of non-smokers to compare the physical performance between smokers and non-smokers. The control group in this study used 0 mg nicotine intake on EC. Due to the relatively low dosage of 3 mg nicotine for both EC and TC (1.5 mg per cigarette), the effects of nicotine on HRR were significantly different after exercise in the TC group but not on SBP and DBP.

In the future, a higher nicotine level for both EC and TC would be recommeded to observe the different physiological responses. A control group should also be used, in which participants would be advised to not smoke before the exercise performance test is conducted.

## Figures and Tables

**Table 1 ijerph-16-01032-t001:** The characteristic data of participants (*n* = 24).

Variable	Mean
Age (year)	23.2 ± 1.7
Height (m)	1.7 ± 0.1
Weight (kg)	63.3 ± 9.3
Body mass index (BMI) (kg/m^2^)	21.8 ± 3.0
Cigarette/day	9.2 ± 1.3
Duration of smoking (year)	3.5 ± 0.8

**Table 2 ijerph-16-01032-t002:** The physiological responses of heart rate (HR) and blood pressure (BP) (*n* = 24).

Variable	C	3EC	3TC
**HR (bpm)**			
Pre-smoking	78 ± 12	79 ± 9	80 ± 11 ^c^
Post-smoking	78 ± 12	80 ± 12	85 ± 11 *
Post-exercise	178 ± 15 ^b,c^	170 ± 18 ^b,c^	168 ± 23 ^b,c^
Change by exercise	+100	+91	+88
**SBP (mmHg)**			
Pre-smoking	117 ± 10	119 ± 9	120 ± 7
Post-smoking	116 ± 12	117 ± 11	121 ± 10
Post-exercise	149 ± 15 ^b,c^	146 ± 16 ^b,c^	145 ± 13 ^b,c^
Change by exercise	+32	+27	+25
**DBP (mmHg)**			
Pre-smoking	76 ± 8	78 ± 9	77 ± 5
Post-smoking	76 ± 10	78 ± 9	83 ± 8 *
Post-exercise	79 ± 9	77 ± 10	77 ± 13
Change by Exercise	+3	−1	0

^b^ Indicates *p* < 0.05 between post-smoking and post-exercise. ^c^ Indicates *p* < 0.05 between pre-smoking and post-exercise. * Indicates *p* < 0.05 from C (Control). Analyzed by two-way ANOVA repeated measures, which were adjusted by the least significant difference (LSD), one-way ANOVA, pair *t*-test and *t*-test, *p* < 0.05. HR: heart rate; SBP: systolic blood pressure; DBP: diastolic blood pressure. C: control (0 mg nicotine of electronic cigarettes), 3EC: 3 mg nicotine of electronic cigarettes, 3TC: 3 mg nicotine of tobacco cigarettes.

**Table 3 ijerph-16-01032-t003:** The exercise performance of the three treatments (*n* = 24).

Variable	C	3EC	3TC
Predicted VO_2max_ (mL/kg/min)	37.7 ± 8.4	37.5 ± 8.7	36.3 ± 8.0
Time to exhaustion (s)	398.5 ± 151.3	399.3 ± 160.7	380.8 ± 149.9

Analysis by one-way ANOVA. C: control (0 mg nicotine of electronic cigarettes), 3EC: 3 mg nicotine of electronic cigarettes, 3TC: 3 mg nicotine of tobacco cigarettes.

**Table 4 ijerph-16-01032-t004:** The Time and frequency domain measures of Heart rate variability (HRV) during exercise (*n* = 11).

Variables	C	3EC	3TC
SDNN (ms)	13.3 ± 4.5	18.2 ± 10.4 *	18.8 ± 12.5 *
RMSSD (ms)	12.8 ± 5.9	19.2 ± 11.9 *	18.8 ± 12.8 *
HR Maximum (1/min)	190.7 ± 13.1	179.8 ± 20.4	173.2 ± 35.1 *
HR Minimum (1/min)	83.4 ± 12.1	84.8 ± 20.0	87.0 ± 13.0
HR Mean (1/min)	151.9 ± 13.4	134.9 ± 27.3 *	140.0 ± 27.4
Peak VLF (ms^2^)	0.035 ± 0.005	0.035 ± 0.005	0.037 ± 0.005
Peak LF (ms^2^)	0.05 ± 0.012	0.06 ± 0.032	0.06 ± 0.028
Peak HF (ms^2^)	0.19 ± 0.044	0.21 ± 0.068	0.23 ± 0.078
Power LF/HF	3.46 ± 3.3	2.43 ± 2.4	2.86 ± 2.0

* Indicates *p* < 0.05 from the C (Control). Analysis by one-way ANOVA and *t*-test one-tailed *p* < 0.05. SDNN: the standard deviation of normal-to-normal (NN) intervals; RMSSD: the square root of the mean of the squared differences between adjacent NN intervals, an estimate of the short-term components of variability; VLF: the very low frequency range, LF: the low frequency; HF: the high frequency.

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
