# Peer review of "Cigarette Smoking Blunts Exercise-Induced Heart Rate Response among Young Adult Male Smokers"

_ijerph, 2019, doi:10.3390/ijerph16061032_

Round 1

Reviewer 1 Report

This paper investigates the effect of cigarette or ENDs use on exercise induced heart rate. The design and statistical analysis appear solid. However, the writing could use improvement.

Overall the quality of writing needs to be improved to bring the paper up to academic standard. The introduction reads like dot points strung together, rather than a cohesive argument for the study objectives. The broader significance of the study is lost. The methods section reads like an undergraduate lab report. attention needs to be paid to sentence and paragraph structure.  

In the discussion - line six of the first paragraph "Previous studies conducted with healthy young adult smokers found that smoking 23 cigarettes pre-exercise.." Is the number 23 a typo? that seems like a lot of cigarettes to smoke before exercise. The following sentence includes the phrase "study proved that...", I would suggest using a difference word than "proved". 

There seems to be inconsistent use of abbreviations with the zero nicotine condition being referred to as "C" and "0EC", it is somewhat confusing. 

Last paragraph before the conclusion (top of page 7) - "Our study reveals that smoking with and without nicotine..." would be clearer if it explicitly stated that the "without nicotine" condition was ENDs.

The participant number is very low, was the study a pilot? Is it considered to be a pilot before conducting a larger study? If so, I would suggest that is stated in the paper. If the study is not a pilot, I am unsure about its validity to contribute to the scientific base of knowledge regarding ENDs.  

Author Response

Dear Reviewer, 

Thank you very much for your comments to improves my study. 

I tried my best to improves the English used MDPI English editing with ID 8231.

The number of 23 cigarettes is a typo. It was 2 - 3 of cigarettes before exercise.

The word "proved" changes to be "reported", and the study use C as a control. 

Smoking with nicotine refers to ENDS and without nicotine ENNDS already mention it. 

Thanks, it is a pilot study to report the effect of acute smoking. 

Thanks very much for your kindness.

Best regards,
Sri Sumartiningsih

Reviewer 2 Report

The manuscript reports an interesting study regarding an important concern. The main limitations of the study have been sufficiently discussed. However, no considerations have been reported about the possible influence of different BMI (ranging from 16.65 to 29.41!) of participants on the results. 

The sentence "Smoking is associated with an increased HR during rest [5]." in the introduction seems a repetition. Please address this issue.

Author Response

Dear Reviewer 2

Thank you very much for your comments to improves the manuscript.

In the study not considered to reported the different of BMI because the treatment was acute smoking with different level of nicotine in the same participant. 

The sentence "Smoking is associated with an increased HR during rest [5]." The sentence before that sentence mention and the same things, now already combined.
Thanks very much for your kindness.

Best regards,
Sri Sumartiningsih
